# Responses of Bacterial Taxonomical Diversity Indicators to Pollutant Loadings in Experimental Wetland Microcosms

**Subhomita Ghosh Roy [1,*], Charles F. Wimpee [1], S. Andrew McGuire [1,2] and Timothy J. Ehlinger [1,2]**

[1]  Department of Biological Sciences, University of Wisconsin-Milwaukee, P.O. Box 413, Milwaukee, WI 53201-0413, USA; cwimpee@uwm.edu (C.F.W.); smcguire@uwm.edu (S.A.M.); ehlinger@uwm.edu (T.J.E.)

[2]  Institute for Systems Change and Peacebuilding, University of Wisconsin-Milwaukee, P.O. Box 413, Milwaukee, WI 53201-0413, USA

[*]  Correspondence: subhomitagr@gmail.com

**Abstract:** Urbanization results in higher stormwater loadings of pollutants such as metals and nutrients into surface waters. This directly impacts organisms in aquatic ecosystems, including microbes. Sediment microbes are known for pollution reduction in the face of contamination, making bacterial communities an important area for bioindicator research. This study explores the pattern of bacterial responses to metal and nutrient pollution loading and seeks to evaluate whether bacterial indicators can be effective as a biomonitoring risk assessment tool for wetland ecosystems. Microcosms were built containing sediments collected from wetlands in the urbanizing Pike River watershed in southeastern Wisconsin, USA, with metals and nutrients added at 7 day intervals. Bacterial DNA was extracted from the microcosm sediments, and taxonomical profiles of bacterial communities were identified up to the genera level by sequencing 16S bacterial rRNA gene (V3–V4 region). Reduction of metals (example: 90% for Pb) and nutrients (example: 98% for $NO3^-$) added in water were observed. The study found correlations between diversity indices of genera with metal and nutrient pollution as well as identified specific genera (including *Fusibacter*, *Aeromonas*, *Arthrobacter*, *Bacillus*, *Bdellovibrio*, and *Chlorobium*) as predictive bioindicators for ecological risk assessment for metal pollution.

**Keywords:** microcosm; constructed wetland; bioindicator; metal; pollutant; ecological risk assessment; water pollution



## 1. Introduction

The root stressors of urbanization impact aquatic ecosystems, including the bacterial community [1–4] (Figure 1). In this context, bioindicators can be very helpful as they can detect signals of pollutant impacts across various temporal and spatial scales and provide a dynamic picture of the environment (Figure 1) [5–7]. Previous studies have identified potential plant bioindicators for pollutants including metals and nutrients [2,8]. Investigating wetland bacterial communities as bioindicators is an emerging field of study to understand the effects of land use change, as well as associated pollutant inputs (nutrients and metals), on the maintenance of ecosystem stability and resilience after contamination [1–3,9–13]. These studies have shown that bacteria found in wetland sediments reduce the levels of pollutants through biogeochemical processes that retain pollutants in the sediments, thus contributing to overall ecosystem health [4,14–17]. Hence, bioindicators such as sediment bacteria are responsive enough in detecting ecological variations in watersheds [18,19] and thus are helpful in conducting sediment risk assessments from pollutants such as metals or nutrients (Figure 1) [7,20,21]. The current study utilized microcosms consisting of sediment collected from constructed wetlands in the Pike River, a rapidly urbanizing watershed in southeastern Wisconsin. These wetlands were constructed for the purpose of stormwater

control [22–24] and receive runoff from a combination of agricultural, commercial, residential, undeveloped, and industrial land uses [2,25]. Hence, it is a suitable site to depict the impact of rapid urbanization (Figure 1). Microcosm systems have been widely used to examine the fate, transport, and treatment of pollutants in a wetland ecosystem [26–28]. An experimental microcosm setup allows for manipulations in treatment, pollutant types, and loading rate that are not possible in natural systems, allowing for the identification of detailed and specific response patterns of bioindicators when exposed to pollutants in a continuous manner [29]. In terms of bacterial indicators, diversity indices are used widely to characterize bacterial communities [26,30] with the identification of specific bacterial taxa as bioindicators within a community, illuminating crucial information about pollutants in an ecosystem [31].

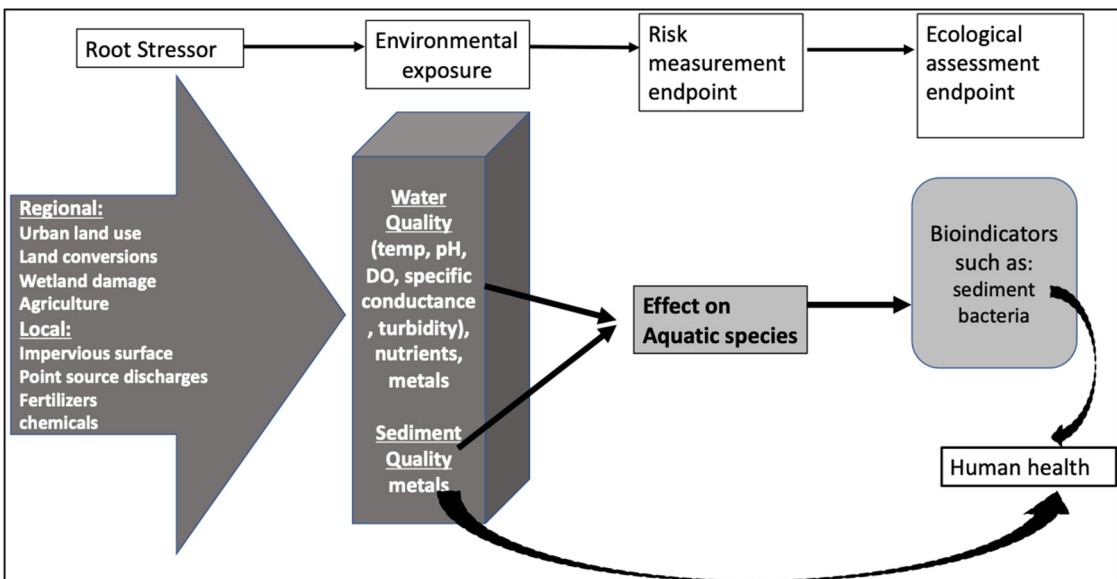

**Figure 1.** Risk propagation model for watershed-based aquatic ecological risk assessment (adapted from [32,33]).

The overarching goal of the study was to determine whether bacterial indicators can serve as a tool for ecological risk assessment of wetland ecosystems (Figure 1), leading to the research questions for this study: (1) Does the response of sediment bacterial indicators (Shannon and Simpson diversity indices of genera) correlate with manipulated changes in pollutant concentration? (2) Can microcosm experiments identify specific assemblages of bacterial taxa that can serve as predictive bioindicators for water pollution?

## 2. Materials and Methods

### 2.1. Sample Collection and Construction of Microcosms

Sediment samples were collected from four wetland sites (1–4) (Figure 2) in the Pike River watershed during summer 2017. These wetlands were constructed for flood mitigation between 2001 and 2008 [22–24], and to the current day receive stormwater runoff from catchments comprised of agricultural, commercial, residential, undeveloped, and industrial land uses (Figure 2, Table 1) [2,8,25]. Hence, it is a suitable site to depict the impact of rapid urbanization.

Two sediments samples were collected per wetland site using an Ekman dredge grab sampler (15 × 15 × 25 cm). Sediment samples were collected and held in 4.73-L plastic closed containers, transported on ice to the laboratory, and stored at 0–2 °C. Microcosms were constructed using similar 4.73 L plastic containers (22 cm diameter × 18 cm height) where sediments were placed to a level of 5 cm from the bottom with reverse osmosis (RO) water added to a level of 10 cm from the bottom [34,35] (Figure 3). Microcosms were placed

in a controlled growth chamber with a temperature between 25 and 30 °C throughout the duration of the study.

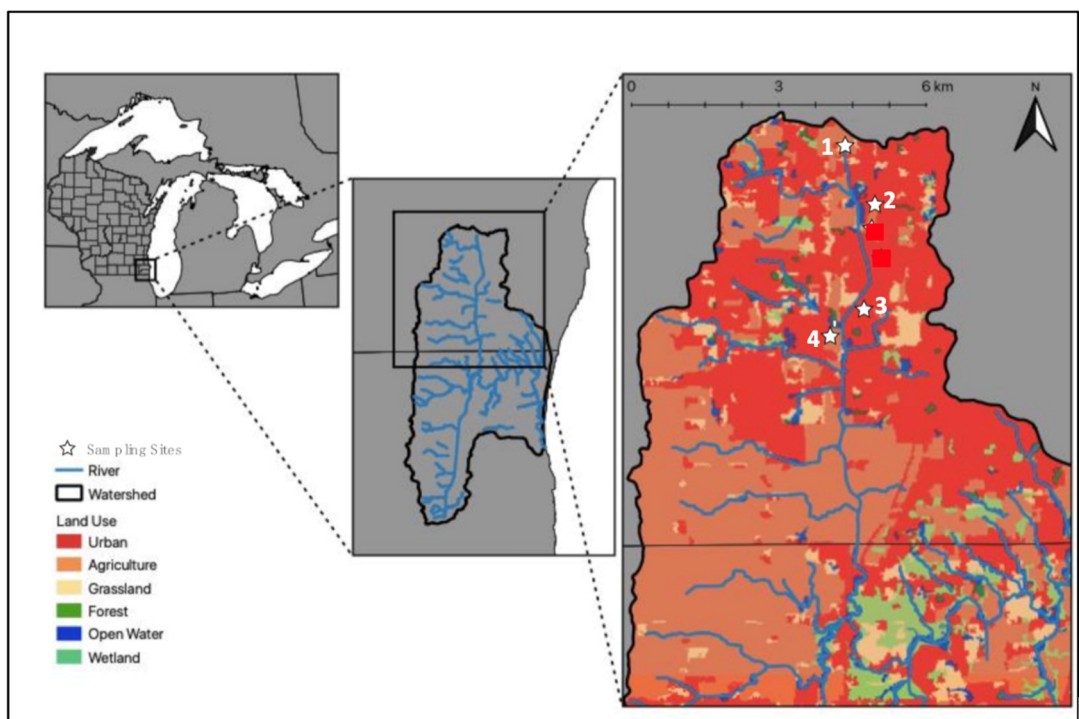

**Figure 2.** Map of the Pike River North Branch (42°43′ N and 87°57′ W) with the surrounding land use (adapted from Southeastern Wisconsin Regional Planning Commission, [25]). Wetland sampling sites are numbered as 1–4 from north to south. This figure is an adaptation from the figure published in [2].

**Table 1.** Wetland site characteristics, wetland sites 1–4 in the Pike River watershed (adapted from [2,25]).

| Land Use Percent in Pike River Watershed | | | | | |
|---|---|---|---|---|---|
| **Wetland Site** | **Watershed Area (ha)** | **Percent Residential** | **Percent Commercial** | **Percent Industrial** | **Percent Agricultural** | **Percent Underdeveloped** |
| 1 | 104.45 | 11 | 15.1 | 12.1 | 61.6 | 0 |
| 2 | 334.18 | 42.3 | 0 | 0 | 57.5 | 0 |
| 3 | 493.72 | 15.7 | 14.2 | 20.08 | 0 | 49.3 |
| 4 | 720 | 0 | 72.2 | 20.2 | 0 | 7.2 |

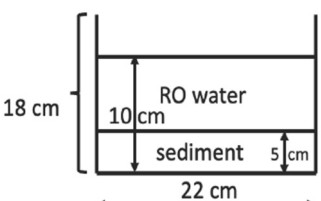

**Figure 3.** Schematic diagram of the microcosms with sediment and reverse osmosis (RO) water and sediment.

## 2.2. Types of Microcosms

For each wetland site, six microcosms were built (24 total). Two treatments microcosms (one high and one low pollutant concentration) were created to examine the effects of metals and nutrients. Each set of treatment microcosms was paired with a control microcosm where no pollutants were added (Figure 4). The nutrient experiments (also called

nutrient treatments or nutrient microcosms) and the metal experiments (also called metal treatments or metal microcosms) both followed similar designs, with pollutants added to the microcosms on day 0 as well as the water changed, and pollutants were re-introduced in 7 day cycles. Microcosm experiments exposed to metal pollution ran for 15 days, while nutrient experiments ran for 30 days.

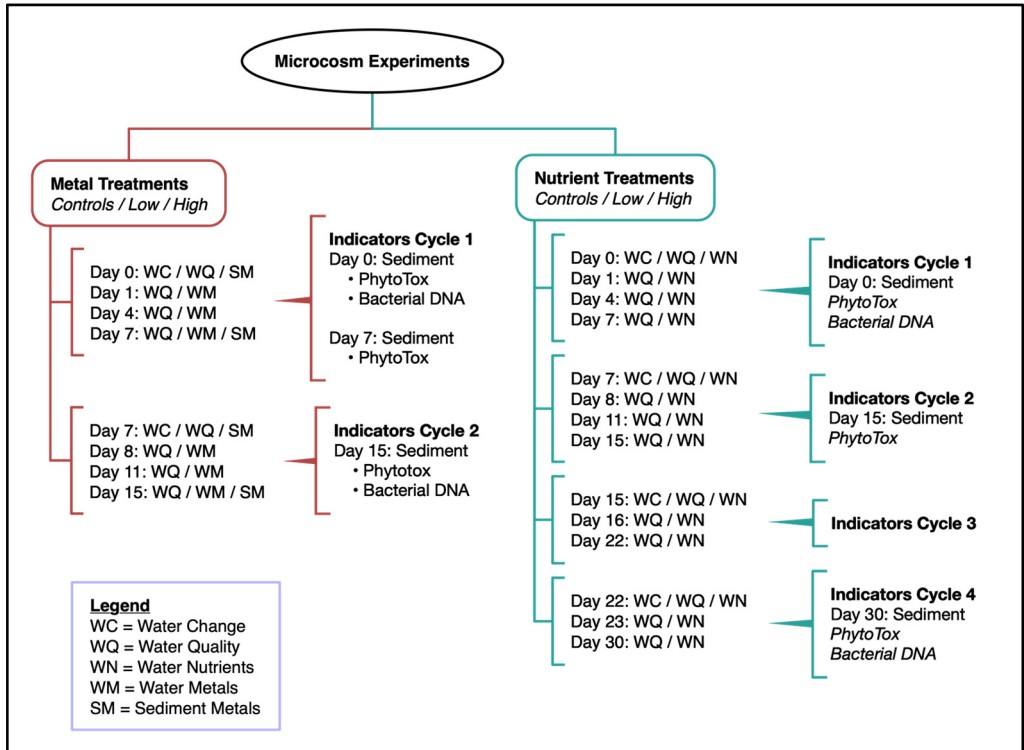

**Figure 4.** Experimental design for wetland sediment microcosm experiments that manipulated loadings of pollutant metals (metal treatments) and pollutant nutrients (nutrient treatments). Controls include sediment without pollutants. In test sediments, pollutants were added in two (low and high) concentrations. This same design was followed to build microcosms with sediments collected from wetland sites 1, 2, 3, and 4.

*2.3. Pollutants*

The wetland site sediments used to construct the microcosms are dominated by urban and agricultural land uses (Figure 2). A previous study detected the presence of nutrients and metals in the water and sediments in these wetland sites [2]. The treatment microcosms were loaded with phosphate ($Na_2HPO_4$) and nitrate ($KNO_3$) for nutrients, and lead ($PbNO_3$) and copper ($CuSO_4$) for metals. Pollutants were dissolved in RO water to produce the desired concentrations (low and high concentrations; see Table 2) and poured over the sediments [27,34–39]. Control microcosms were constructed with wetland sediment and RO water and ran the course of the experiment without pollutants added.

**Table 2.** Metal and nutrient microcosm experiment design with sediment treatment type for each wetland (1, 2, 3, and 4) in control and test sediments.

| (A) Metal Microcosm | | | |
|---|---|---|---|
| | **Test Sediment** | | |
| **Metal Added to Water** | **Control** | **Low Concentration** | **High Concentration** |
| Cu as $CuSO_4$ | 0 mg/L | 0.05 mg/L | 0.15 mg/L |
| Pb as $PbNO_3$ | 0 mg/L | 0.1 mg/L | 0.3 mg/L |

**Table 2.** *Cont.*

| (B) Nutrient Microcosm | | | |
| --- | --- | --- | --- |
| | Test Sediment | | |
| Nutrient Added to Water | Control | Low Concentration | High Concentration |
| $NO_3^-$ as $KNO_3$ | 0 mg/L | 5.0 mg/L | 15.0 mg/L |
| $PO_4^{3-}$ as $Na_2HPO_4$ | 0 mg/L | 1.0 mg/L | 3.0 mg/L |

*2.4. Measurements in Microcosm*

Both high and low concentrations of metals (Pb and Cu) and nutrients ($NO_3^-$ and $PO_4^{3-}$) were added to the microcosms at day 0 (Figure 4, Table 2). Pollutant concentration in microcosm water was measured along with water quality characteristics (pH, temperature, dissolved oxygen, specific conductance, and turbidity) on days 1, 4, 7, 8, 11, and 15 for metal microcosms and days 1, 4, 7, 8, 11, 15, 16, 22, 23, and 30 for nutrient microcosms (Figure 4). In all treatment microcosms, the water was decanted down to the level of the sediment layer and replaced with water containing initial treatment concentrations of pollutants at 7 day intervals (once in metal microcosms, twice in nutrient microcosms). The 7 day interval was selected on the basis of the recommended retention time for artificial wetlands (marshes and ponds) in wastewater treatment systems (4–12 days) from the United States Environmental Protection Agency [40].

Within the metal microcosms, concentrations of Pb and Cu in water (WM in Figure 4) were measured using inductively coupled plasma mass spectrophotometry (ICP-MS). Analysis was conducted at the University of Wisconsin-Milwaukee School of Freshwater Sciences utilizing a Thermo Scientific Element 2 High Resolution Sector field ICP-MS [41]. In the nutrient microcosms, $NO_3^-$ and $PO_4^{3-}$ in water were analyzed with an HACH DR 2800™ spectrophotometer using powder pillow test kits—cadmium reduction method for $NO_3^-$ (in mg/L) and ascorbic acid method for $PO_4^{3-}$ (in mg/L). Water quality parameters in all microcosms were measured using YSI 6600 EDS™ multi-parameter sondes [2,42].

In addition, sediment samples were collected on days 0 and 15 from metal microcosms and on days 0 and 30 from nutrient microcosms to identify bacterial taxonomical diversity of genera (Figure 4). The bacterial DNA was extracted using a DNA™ spin kit for soil [2,43–45].

To measure concentrations of metals in the sediments (SM in Figure 4), we collected samples on days 0, 7, and 15 and analyzed them using X-ray fluorescence (XRF) [2]. Due to constraints in sediment amount, for the day 0 sediment measurements, aggregate sediment samples by wetland sites were used before dividing among the microcosms.

*2.5. Data Analysis*

Distributional properties were examined for all data collected from the microcosm experiments, and all statistical analyses were conducted using JMP™ Software Version 14.0 [46]. Contrasts in pollutant concentrations among treatment levels and the experimental timeline for water nutrients (WN), water metals (WM), and sediment metals (SM) (Figure 4) were calculated for each treatment using multifactor analysis of variance (ANOVA). The effects of treatment level and experimental timeline on bacterial indicators were also examined using multifactor analysis of variance (ANOVA).

Forward stepping multiple regression was used to determine best-fit models for the predictive linear relationships between concentrations of pollutants added to the microcosms (metals and nutrients) and bacterial community indicators (Shannon and Simpson diversity indices of genera).

The taxonomical profile of the sediment bacterial communities was determined from sequencing of the 16S bacterial rRNA (v3–v4 region) [2] identified to the genus level. The sequences were retrieved electronically. The bioinformatics analyses were performed with the software Mothur (v1.36.1). This analysis used the SILVA database (Release Version 128) for sequence alignment and Greengenes Reference Taxonomy (Version13_8_99) for taxonomy [2,47].

Hierarchical cluster analysis was performed with the identified communities in each wetland site for each type of microcosm (metal and nutrient). On the basis of visual examination of change in abundance within the clusters throughout the experiments relative to their response to the pollutants, we identified the bacterial genera and categorized them as intolerant, sensitive, and tolerant.

## 3. Results

### 3.1. Time and Treatment Effects on Metals and Nutrients in Microcosms

The study aimed to find out if bacterial indicators can be a tool for ecological risk assessment for wetland ecosystems. The literature has shown that bacteria found in wetland sediments can decrease pollutant levels [14–17]. Assuming the bacterial communities present in the microcosms will help in the reduction. The first analysis of the study was conducted to identify any pollutant reduction in the microcosms across the experimental timeline. Effects of treatment level on concentrations of metals (in the water and sediments) and nutrients (in water) across time are shown in Appendix A Table A1.

In both low and high treatment levels, metal microcosms showed a significant reduction in Pb concentrations in the microcosm water ($p$-value < 0.0001) (Figure 5 and Appendix A Table A1). The low treatment microcosm's concentrations of Pb were 0.00409 ppm and 0.00037 ppm on days 0 and 15, respectively (90% reduction). The high treatment microcosm's concentrations of Pb were 0.00575 ppm and 0.00038 ppm on days 0 and 15, respectively (93% reduction) (Figure 5).

In sediments, As decreased from 0.87 ppm at day 0 (control) to 0.35 ppm in low (59% reduction) and 0.31 ppm in the high (64% reduction) treatment microcosms (Figure 6 and Appendix A Table A1). However, the Pb detected in the microcosm showed a different story. The Pb in the microcosm increased throughout the experiment from 0.002 ppm at day 0 (control) to 0.004 ppm in low (100% increase) and 0.008 ppm in high (300% increase) treatments. None of the other metals from sediment (Ag, Cd, Fe, Hg, Ni, Rb, Zn) showed any significant changes (Figure 6 and Appendix A Table A1).

Nutrient concentration in microcosm water decreased over the course of the experiment for both low and high treatments of $NO_3^-$ ($p$-value = 0.0036) and $PO_4^{3-}$ ($p$-value < 0.0001) (Figure 5). $NO_3^-$ concentration decreased from 7.28 mg/L at day 0 to 0.14 mg/L (98% reduction) for the low treatment and 0.87 mg/L to 0.86 mg/L (1.14% reduction) for the high treatment. $PO_4^{3-}$ concentration decreased from 1.12 mg/L at day 0 to 0.5 mg/L (55% reduction) for the low treatment and 1.635 mg/L to 0.73 mg/L (55% reduction) for the high treatment. The effect of treatment level was not significant for Pb, Cu, $NO_3^-$, and $PO_4^{3-}$ (Figure 5 and Appendix A Table A1).

### 3.2. Time and Treatment Effects on Bacterial Bioindicators

The sequence alignment and taxonomical analysis of the extracted bacterial DNA sequences were conducted using SILVA database Release Version 128 and Greengenes Reference Taxonomy Version 13_8_99 [47]. Taxonomic profiles were determined up to the genus level of classification for each microcosm. Among a total of 175,207 sequences, 70 unique phyla, and 32,848 unique genera were identified. A list of the major phyla is listed in Appendix A Table A2. Shannon and Simpson diversity indices of genera were used to calculate the bacterial indicators on the basis of the unique genera identified across both nutrient and metal microcosms (Figure 7).

Bioindicators of wetland sediments have been observed to be sensitive and responsive in detecting ecological changes in watersheds [18,19]. In both low and high treatment metal microcosms, the Simpson diversity indices of genera decreased significantly throughout the experiments compared to the control ($p$-value = 0.0314). On day 0, Simpson diversity index was measured to be 74 and at day 15 decreased to 69 in the low (6.75% reduction) and 41 in the high (44.59% reduction) treatment microcosms (Figure 7 and Appendix A Table A3).

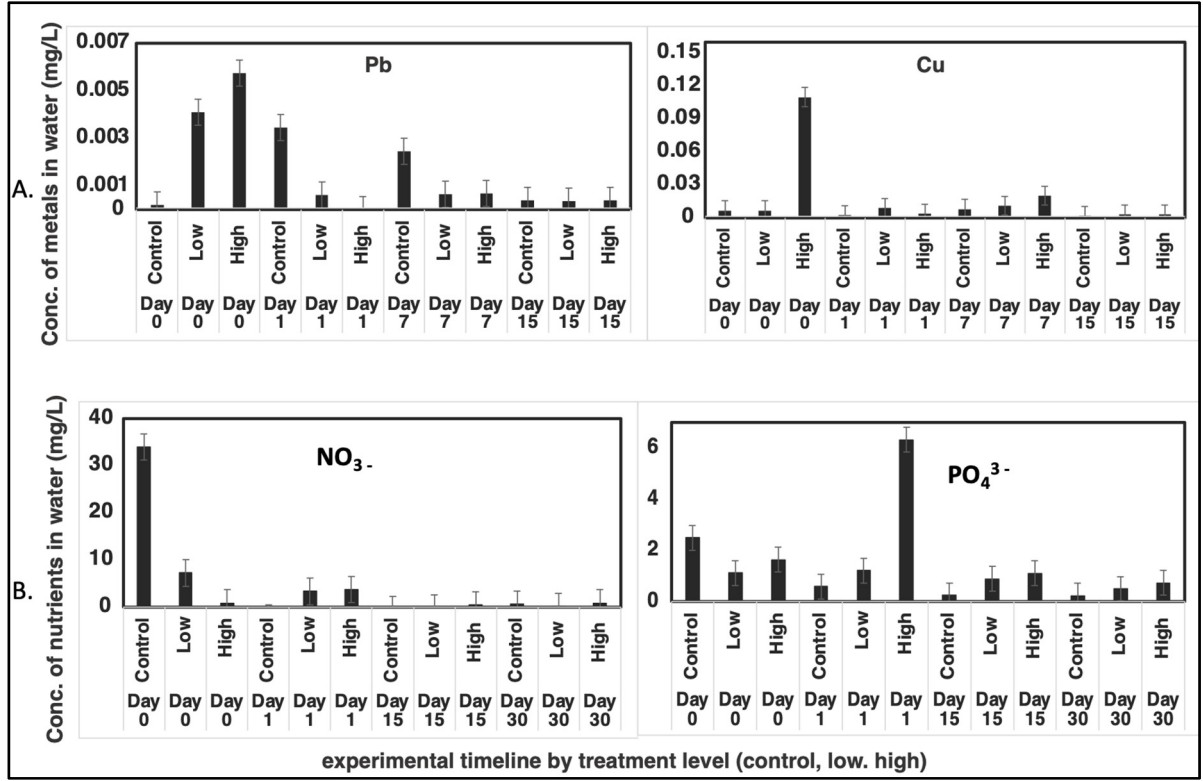

**Figure 5.** (**A**) Median concentration of Pb and Cu (mg/L) in metal microcosms (for wetland sites 1–4) at days 0, 1, 7, and 15 of the experiment. (**B**) Median concentration of $NO_3^-$ and $PO_4^{3-}$ (mg/L) in nutrient microcosms (for wetland sites 1–4) at days 0, 1, 15, and 30 of the experiment. This test included a control treatment where no pollutants were added, accompanied by sets of low and high concentrations in the treatment microcosms, where metals (Pb and Cu) and nutrients ($NO_3^-$ and $PO_4^{3-}$) were added to the water as per the experimental design in Table 2. Bars show mean $\pm$ 1 SE. Multifactor ANOVA tables are in Appendix A Table A1.

In the nutrient microcosms, the Shannon (*p*-value = 0.0013) and Simpson (*p*-value = 0.0418) diversity indices of genera decreased significantly throughout the experiment. At day 0, Shannon diversity was measured to be 4.95 and decreased to 4.85 at the low (2.02% reduction) and 4.90 (1.01% reduction) at the high treatment microcosms (Figure 7 and Appendix A Table A3). For nutrient microcosms, the Simpson diversity index decreased from 78 at day 0 to 60 (23 % reduction) in the low and 70 (10.25% reduction) in the high treatment microcosms at day 15. However, these decreases were statistically significant compared to the control (Figure 7 and Appendix A Table A3).

### 3.3. Effects of Pollutants on Bacterial Bioindicators

In the next part, relationships between pollutants and the bacterial bioindicators were examined. Forward-stepping multiple regression analyses found significant, predictive relationships between the bioindicators and metal pollutant levels measured in the water (Table 3). The effect of increasing Cu concentration in the water of metal microcosms significantly decreased the Shannon and Simpson diversity indices of genera (Table 3; Figure 8). Higher Pb concentration caused a significant increase in the Shannon and Simpson diversity indices of genera (Table 3; Figure 8). For sediment metals in metal microcosms, Zn concentration positively correlated with increased Simpson diversity of genera (Figure 8). There were no significant relationships between nutrient loading in the nutrient microcosms and bacterial bioindicators.

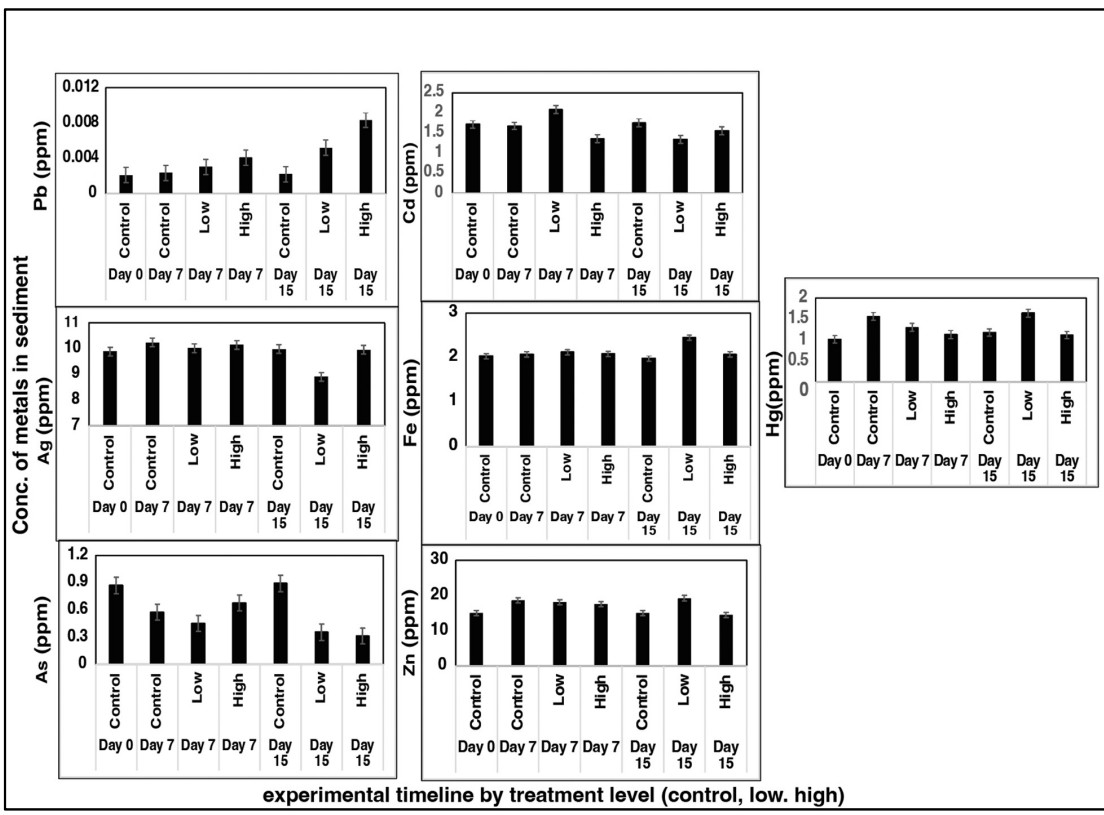

**Figure 6.** Median concentration of (in ppm) detected sediment metals in metal microcosms (for wetland sites 1–4) at days 0, 7, and 15 of the experiments. This test included a control treatment where no pollutants were added, accompanied by sets of low and high concentrations in the treatment microcosms, where metals (Pb and Cu) were added to the water as per the experimental design in Table 2. Bars show mean ± 1 SE. Multifactor ANOVA tables are in Appendix A Table A1.

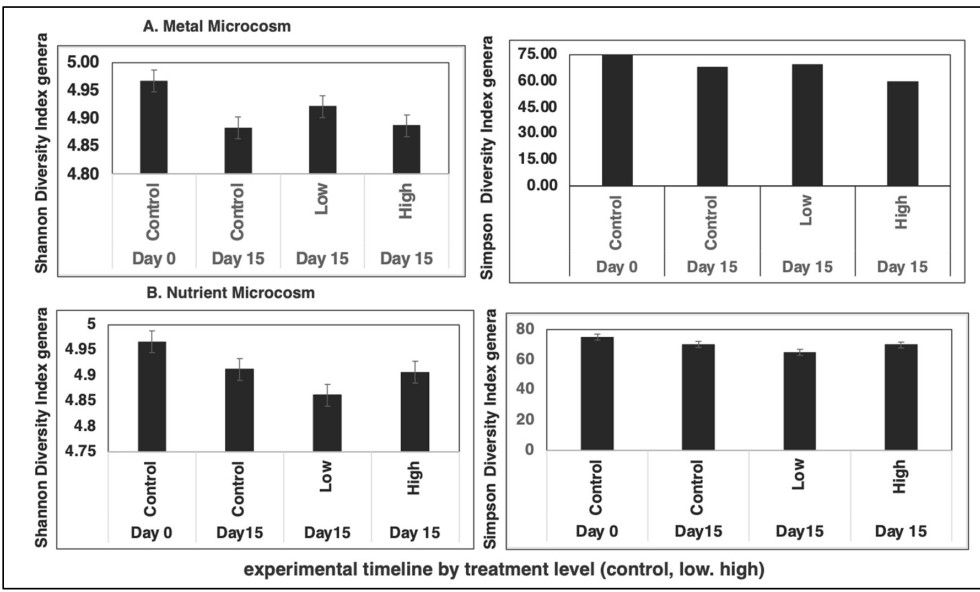

**Figure 7.** Shannon and Simpson diversity indices of genera during at days 0 and day 15 of metal (**A**) and nutrient (**B**) microcosm experiments. This test included a control treatment where no pollutants were added, accompanied by sets of low and high concentrations in the treatment microcosms, where metals (Pb and Cu) and nutrients ($NO_3^-$ and $PO_4^{3-}$) were added to the water as per the experimental design in Table 2. Bars show mean ± 1 SE. Multifactor ANOVA results are presented in Appendix A Table A3.

**Table 3.** Multiple regression model with estimate (slope), *p*-value (significance), and combined $R^2$ of the relationship between Shannon and Simpson diversity indices of genera and the concentration of Pb and Cu added (mg/L) to the water of metal microcosms. * in the *p*-value column indicates a strong probability of significance.

| Parameter | Genera Shannon Diversity Index | | | Genera Simpson Diversity Index | | |
|---|---|---|---|---|---|---|
| | Estimate | *p*-Value | $R^2$ | Estimate | *p*-Value | $R^2$ |
| Concentration of Pb (mg/L) in water | 254 | <0.0001 * | | 10187 | <0.0001 * | |
| Concentration of Cu (mg/L) in water | −116 | <0.0001 * | 0.64 | 1797 | <0.0001 * | 0.64 |

### 3.4. Predictive Indicator Categories Based on Key Pollutants in Field and Microcosm Study

In the last step of the analysis, specific bacterial genera were identified in relation to the responses of the bacterial indicators with the metal pollutants (Pb and Cu). The results above indicate that metals added to the microcosms have a significant effect on the sediment bacterial community. To depict the abundance level of each genus identified relative to their response to metals added, we performed a hierarchical cluster analysis in each metal microcosm (Figures 9 and 10). On the basis of these clusters, we divided the identified bacterial genera into three categories:

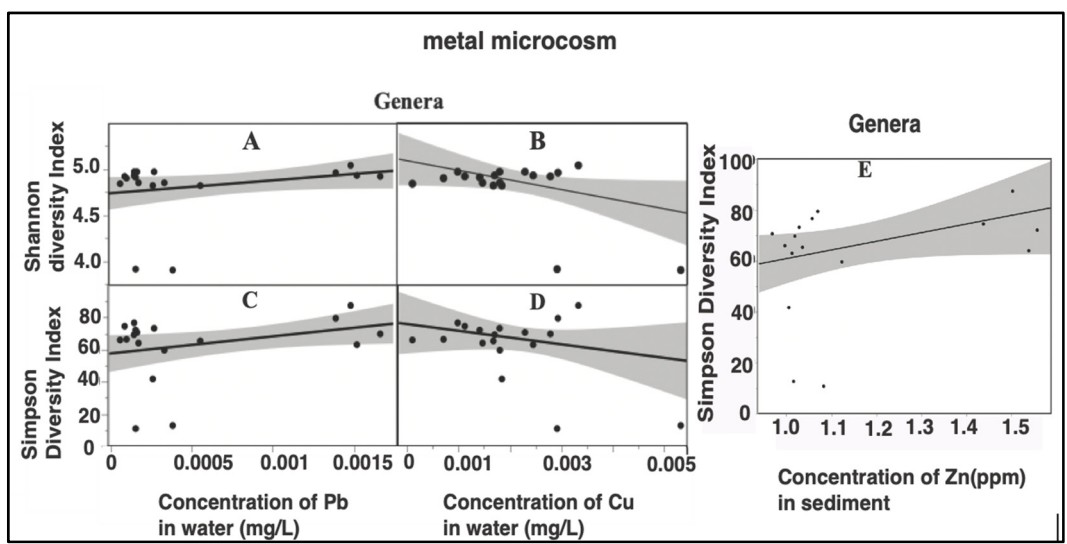

**Figure 8.** Relationship of the Shannon and Simpson diversity indices of genera with increasing concentrations of Pb (**A**,**C**) and Cu (**B**,**D**) in microcosm water (mg/L), along with the relationship between the Simpson diversity index of genera and increasing concentrations of Zn (**E**) in the sediment (ppm). All other statistically non-significant relationship between detected sediment metals in the metal microcosm and diversity indices of genera are listed in Appendix A Table A4.

Intolerant bacterial genera: genera that were present at the start of an experiment but disappeared later in both the low and high treatments.

Sensitive bacterial genera: genera that were present in the low treatment but absent in the high treatment of the metal microcosm near the end of the experiment.

Tolerant bacterial genera: genera present at the start and end of the low and high treatments.

Additional sub-types of tolerant genera were also observed and were designated as less tolerant (genera present at the start and end of the low treatments) and highly tolerant (genera present at the start and end of the low and high treatments).

Some bacterial genera not found at the start of the experiment appeared later. These genera may have been introduced by contamination or were simply not detected at the start of the experiment. Due to this inconsistency, these genera were not included in the

analysis. The detected genera among all the wetland site microcosms are listed in Table 4 by their category (intolerant, sensitive, and tolerant).

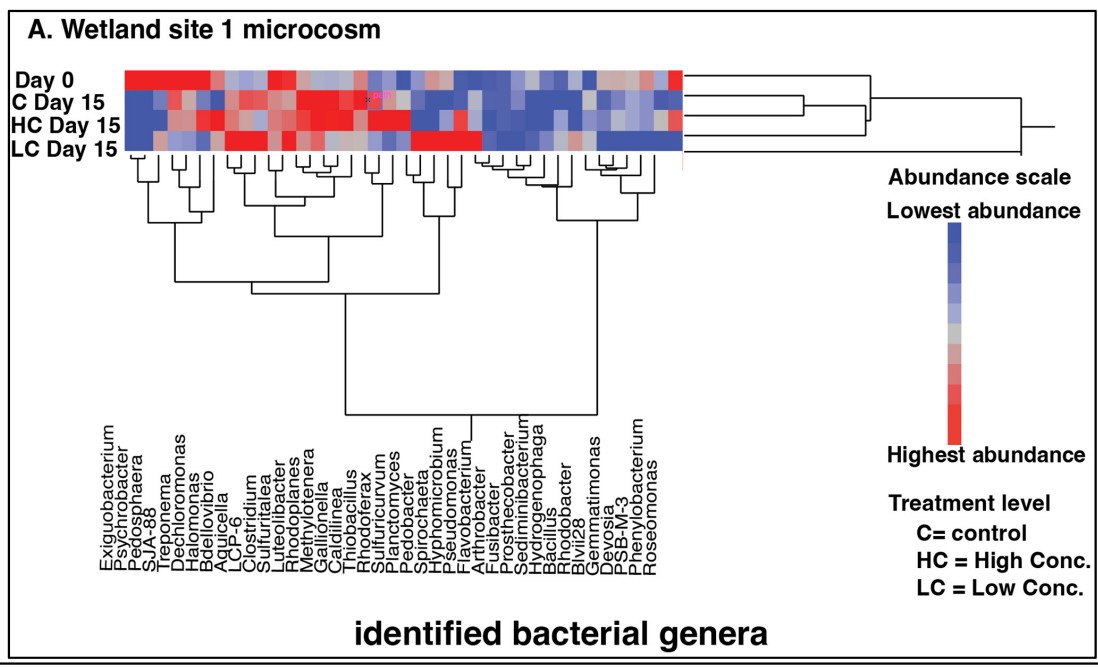

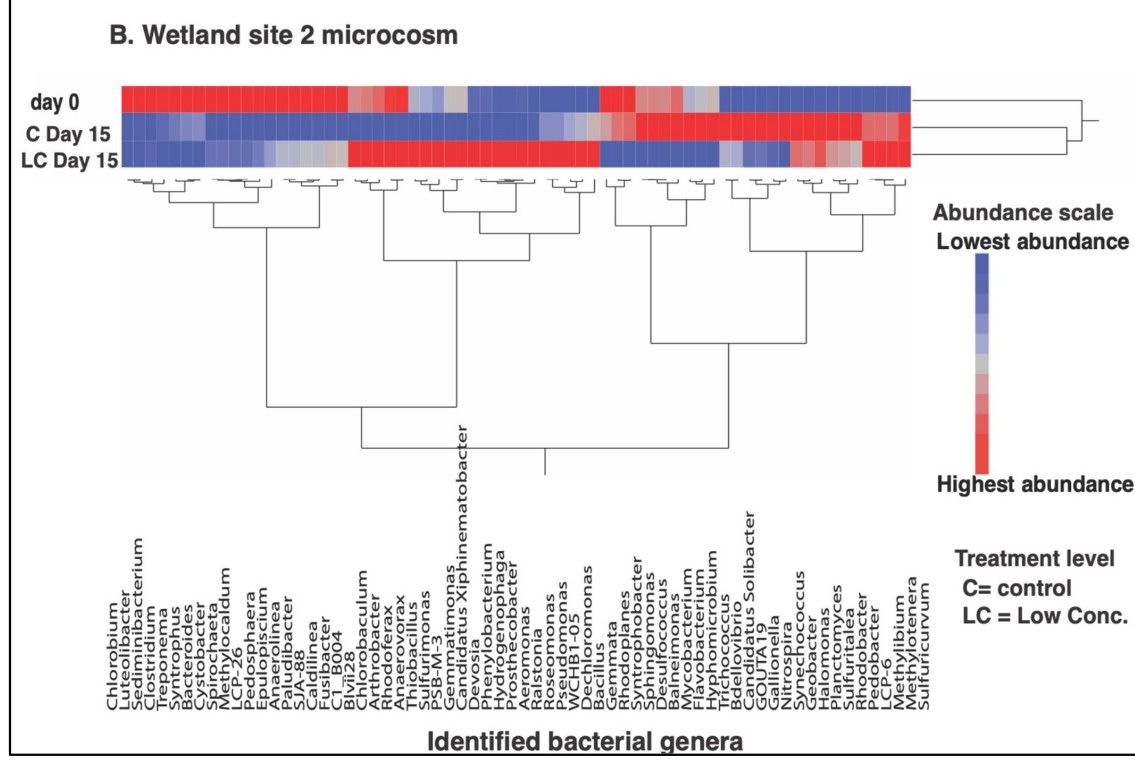

**Figure 9.** Hierarchical cluster analysis of the bacterial genera detected in the sediments of the treatments of wetland site 1 (**A**) and 2 (**B**) metal microcosm, where day 0 = beginning treatment of the metal microcosm experiment; C Day 15 = treatment of the metal microcosm experiment of the native control sediment with no metals added; LC Day 15 = end treatment (day 15) of the metal microcosm experiment with metals (Pb and Cu) added in lower concentrations in the water to the sediment; HC Day 15 = end treatment (day 15) of the metal microcosm experiment with metals (Pb and Cu) added in higher concentrations in the water to the sediment, as illustrated in the experimental design under Table 2.

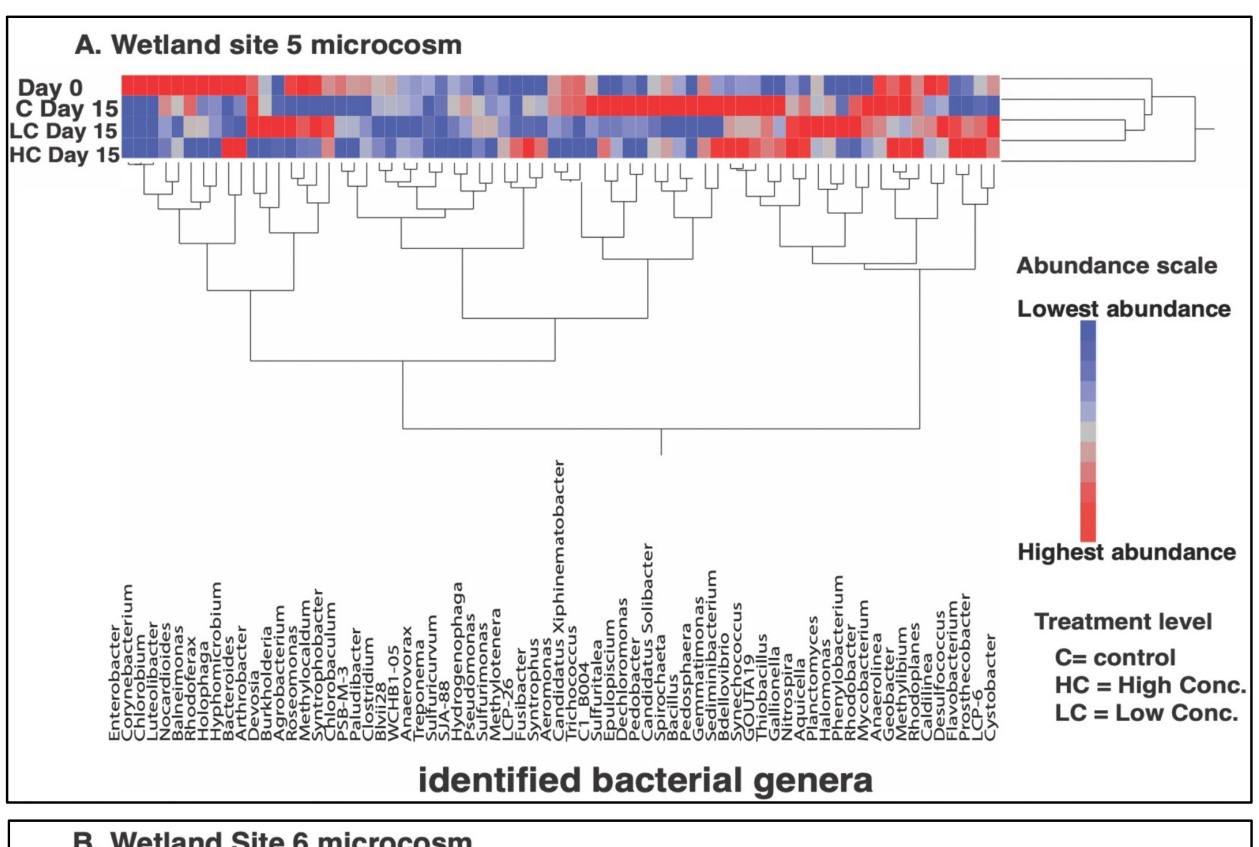

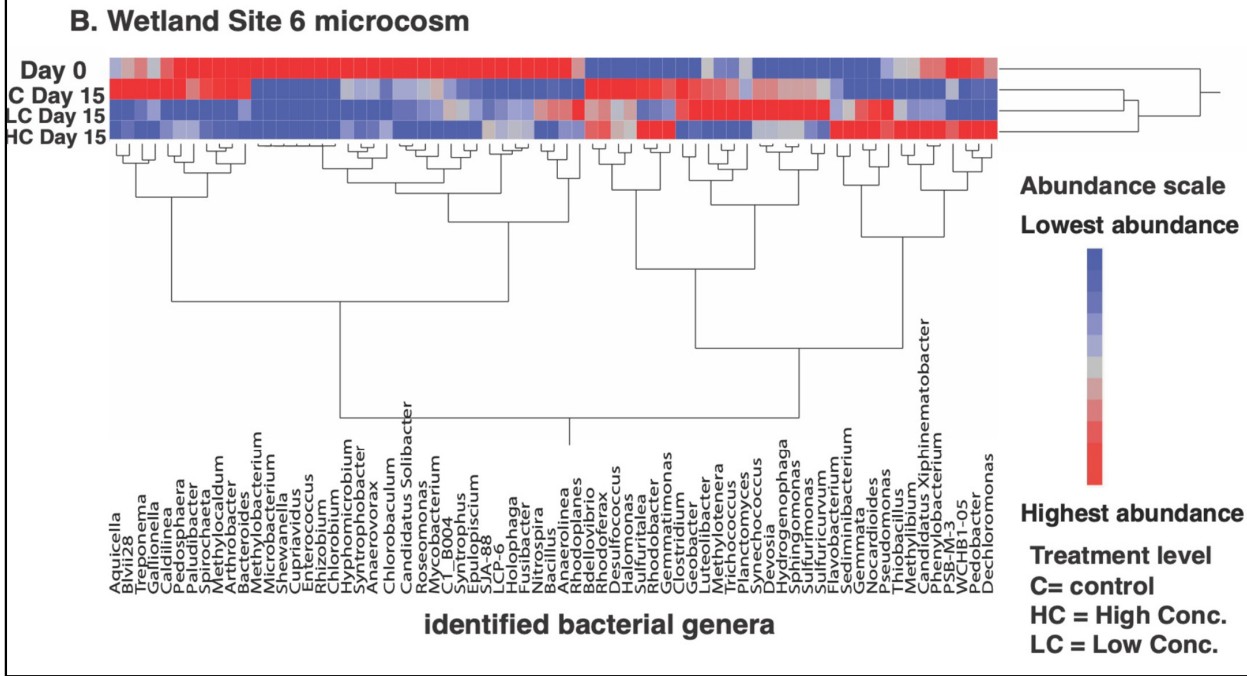

**Figure 10.** Hierarchical cluster analysis of the bacterial genera detected in the sediments of the treatments of wetland sites 3 (**A**) and 4 (**B**) metal microcosm, where day 0 = beginning treatment of the metal microcosm experiment; C Day 15 = treatment of the metal microcosm experiment of the native control sediment with no metals added; LC Day 15 = end treatment (day 15) of the metal microcosm experiment with metals (Pb and Cu) added in lower concentrations in the water to the sediment; HC Day 15 = end treatment (day 15) of the metal microcosm experiment with metals (Pb and Cu) added in higher concentrations in the water to the sediment, as illustrated in the experimental design under Table 2.

**Table 4.** Categories of bacterial indicator genera (sensitive, tolerant, intolerant) in the metal microcosms of wetland sites 1, 2, 3, and 4.

| Genera Detected in Metal Microcosm | | | |
|---|---|---|---|
| **Intolerant** | **Sensitive** | **Tolerant** | |
| | | Less tolerant | Highly tolerant |
| *Methylobacterium, Microbacterium, Shewanella, Cupriavidus, Enterococcus, Rhizobium, Enterobacter, Corynebacterium, Chlorobium, Exiguobacterium, Psychrobacter* | *Aquicella, Agrobacterium, Gemmata, Sphingomonas, Nitrospira, Balneimonas, Chlorobaculum, Aeromonas, Pedosphaera, Arthrobacter* | *Aquicella, Candidatus Solibacter, Burkholderia, GOUTA19, Holophaga, Sphingomonas, Syntrophus, Anaerolinea, Geobacter, Cystobacter, Desulfococcus, Syntrophobacter, Paludibacter, Nitrospira, Methylocaldum, WCHB1-05, C1_B004, Candidatus_Xiphinematobacter, Synechococcus, Balneimonas, Trichococcus, Mycobacterium, Sulfurimonas, Epulopiscium, Ralstonia, Pedosphaera, Bdellovibrio, Flavobacterium, Clostridium, SJA-88, Luteolibacter, Treponema, Caldilinea, LCP-6, Thiobacillus, Rhodoplanes, Sulfuritalea, Rhodoferax, Methylotenera, Planctomyces, Rhodobacter, Spirochaeta, Halomonas, Gallionella, Roseomonas, PSB-M-3, Dechloromonas, Bacillus, Devosia, Hydrogenophaga, Sediminibacterium, Phenylobacterium, Sulfuricurvum, Pedobacter, Hyphomicrobium, Pseudomonas* | *Nocardioides, Holophaga, Gemmata, Syntrophus, Bacteroides, Anaerolinea, Geobacter, Candidatus_Solibacter, Cystobacter, Desulfococcu, Syntrophobacter, GOUTA9, Paludibacter, Nitrospira, Clostridium, Blvii28, Methylocaldum, Synechococcus, Candidatus_Xiphinematobacter, WCHB1-05, Aeromonas, Luteolibacter, C1_B004, Methylibium, Anaerovorax, Treponema, Sulfurimonas, Chlorobaculum, Epulopiscium, Caldilinea, Pedosphaera, Bdellovibrio, Flavobacterium, SJA-88, Blvii28, LCP-6, Thiobacillus, Rhodoplanes, Sulfuritalea, Rhodoferax, Methylotenera, Rhodobacter, Gemmatimonas, Spirochaeta, Halomonas, Gallionella, Roseomonas, PSB-M-3, Fusibacter, Dechloromonas, Bacillus, Devosia, Sediminibacterium, Hydrogenophaga, Prosthecobacter, Phenylobacterium, Sulfuricurvum, Arthrobacter, Pedobacter, Hyphomicrobium, Pseudomonas, Ralstonia* |

## 4. Discussion

Root stressors of urbanization, such as land-use change, contribute to pollutant accumulation in aquatic ecosystems and causes environmental hazards (Figure 1) [48–50]. Studies in urbanized intertidal sediments in China [48] and wetlands in India [49] have shown how pollutants such as metals and nutrients can accumulate in an ecosystem. In order for the risks imposed by these accumulated pollutants to be understood, it is crucial to identify responsive bioindicators able to detect ecological effects [18,19]. In this regard, sediment bacteria are an emerging bioindicator as their communities are highly impacted by and are resilient to pollutants. Both these criteria are important in order to be a bioindicator [6,14–17]. The current study aimed to establish bacterial bioindicators as a risk assessment tool by investigating correlations between wetland sediment bacterial diversity with metal and nutrient pollutant concentrations and then identify specific bacterial genera from the wetlands as predictive bioindicators.

### 4.1. Time and Treatment Effects on Pollutants in Microcosms

In the first step of analysis reduction in the pollutants being added to the microcosms across the experimental timeline was examined. It was observed that in the metal microcosms, concentrations of Pb in water decreased significantly over time both in high and low levels of treatment of the microcosm water (Figure 5), but an increase in Pb (in ppm) in the sediments over time was also observed (Figure 5). This implies that the added Pb was removed from the water column and accumulated in the sediments of the microcosms. Aquatic ecosystems, such as wetlands, are known to retain metals such as Pb, forming metal complexes, although this property of metal retention varies with sediment properties such as adsorption and desorption [51]. A study in Savannah River site, Aiken, SC, USA, examined contamination retention in constructed wetland sediment in the form of metal retention (or recalcitrant factor). For Pb, the recalcitrant factor was 73%, indicating Pb can bound to sediment strongly as Pb has an affinity for sulfide, thus forming strong complexes [52]. In similar sediment studies for trace elements in Songkhla Lake, Thailand [51], and Tablas de Daimiel wetland in Spain [53], the researchers also observed accumulation of Pb, often in high concentrations [53], in sediment, and it was associated with reduced sulfur fractions [51].

In the nutrient microcosms, the concentration of $NO_3^-$ and $PO_4^{3-}$ decreased significantly over time both in high and low levels of treatment (Figure 5; Appendix A Table A1). Previous studies have also shown wetlands treat water by removing these pollutants from nutrient-rich waters by mechanisms such as bioretention [54–57]. However, the bioretention varies with the media (sediment composition) [57].

### 4.2. Bacterial Bioindicators and Effect of Pollutants

The study examined the suitability of "bacterial bioindicators" for ecological risk assessment in wetlands. Variation in the bacterial bioindicators (or the diversity indices) across the experimental timeline were analyzed. In metal microcosms, the diversity indices decreased over time (Figure 7). Bioindicators of wetland sediments are known to be highly responsive to detecting ecological changes within watersheds [18,19], as observed for the diversity indices of this experiment.

On the other hand, in relation to the specific pollutants, increasing Cu concentration in the water of metal microcosms significantly decreased the Shannon and Simpson diversity indices of genera (Figure 8; Table 3). Studies suggest that long-term exposure to heavy metals such as Pb, Cu, Cd, and Zn can decrease the microbial biomass, activity, and diversity [58]. A study in Ain River sediment, France, based on 16SrRNA genes quantification, revealed that bacterial community structures showed a clear shift after Cu exposure [59]. Specifically, metals such as Cu (alone or in combination) affect the genetic structure of the exposed bacterial community [59]. However, with increasing Pb concentrations in water of the microcosm, there is a significant increase in both diversity indices (Figure 7; Table 3). In addition, the detected metals in the sediments showed a similar relationship with Pb in water, with higher Zn levels in sediments being correlated with higher Simpson diversity indices of genera (Figure 8). Several studies have demonstrated a change in the microbial community structure and function after exposure to metal pollution [60–62]. Heavy metal pollution has also been shown to create selective pressure on bacterial communities [63,64]. Bacterial communities adapt with metal resistance genes becoming stably present, resulting in their continuing presence even after long-term exposure to metal pollutants [2,60–62,65,66]. For example, a study in Lake DePue, IL, USA, demonstrated that metal pollution impacted microbial community structure and increased the abundance only of certain metal resistance genes [61]. Certain evolutionary processes work behind this selective pressure. Rensing et al. (2002) [67] suggest that mechanisms such as lateral gene transfer (LGT) are the primary active evolutionary process by which soil or sediment microbial communities adapt, which in turn impacts the diversity of a bacterial community as certain genes increase in frequency [60].

In nutrient microcosms, the Shannon diversity index was higher at the start of the experiments but decreased over time (Figure 7; Appendix A Table A3). This is consistent with other studies that showed phosphorus deficiency can negatively affect the growth and development of microorganisms, thus reducing their number and diversity [68].

Overall, the bacterial diversity bioindicators were highly correlated with the manipulated changes of pollutant loadings in the microcosms systems.

### 4.3. Finding of Bacterial Bioindicators

A bioindicator is a single or group of species whose status, functional abilities, or population can represent a picture of the quality of the environment and the cumulative effects of several pollutants present [6]. Soil microbial communities provide a multitude of ecosystem services and thus play an important role in preserving ecosystem function. The results of this study showed that metals have a significant impact on the sediment bacterial community and specific bacterial genera concerning the metal pollutants were identified and categorized (Table 4; Figures 9 and 10).

However, some genera were found to overlap among categories. For example, *Gemmata* and *Sphingomonas* were observed to be sensitive in the wetland 2 microcosms but tolerant in the wetland 4 microcosms. This implies that there was a difference in the sediment chemistry across wetland sites in the watershed (Figure 2). However, some genera had a very clear response trend. For example, *Chlorobium* was observed to be intolerant to the exposure of heavy metals such as Pb and Cu in the microcosms (Figures 9 and 10; Table 4. This genus disappeared in the treatment microcosms relative to the controls during the experiment. Some genera such as *Fusibacter*, *Chlorobaculum*, *Prosthecobacter*, *Nocardioides*, *Aeromonas*, and *Arthrobacter* were observed to be highly tolerant (or resistant) to the exposure of heavy metals such as Pb and Cu in the microcosms (Figures 9 and 10; Table 4). These genera were present throughout the experimental time in response to Pb and Cu (in both low and high treatments of the microcosms) relative to the controls (Figures 9 and 10; Table 4).

Bacterial genera such as *Ralstonia*, *Pseudomonas*, *Flavobacterium*, *Clostridium*, *Bacillus*, *Pedosphaera*, *Bdellovibrio*, *Holophaga*, and *Geobacter* were observed to be tolerant (Figures 9 and 10; Table 4). In some microcosms, these genera were less tolerant, and in some were highly tolerant. As discussed before, this could have been due to the difference in the sediment chemistry as the microcosms were built with sediments collected from different wetlands. This also might be due to the fact possibly these genera belong to different species or strains; hence, their tolerance level is different.

Among the tolerant genera identified, *Ralstonia*, *Flavobacterium*, *Bacillus*, *Pseudomonas*, *Clostrodium*, *Aeromonas*, and *Arthrobacter* have been identified as Pb-resistant, remediating, precipitating, and biomethylating, as well as Hg-resistant and bioremediating [69–76]. Previous studies have also identified *Clostridium*, *Pseudomonas*, *Bacillus*, and *Arthrobacter* as Cu-resistant bacterial genera [77–82]. Bioindicators need to be responsive to a wide range of stresses and be able to discriminate between anthropogenic changes and natural variation [6]. In this study, a wide range of specific soil bacterial bioindicators were chosen in relation to Pb and Cu exposure. Hence, these identified metal-resistant genera can be used as ex-ante impact indicators for ecological risk assessment or biomonitoring tools in constructed wetland ecosystems.

A limitation to this study is that we were only able to identify taxa up to the genus level. In order for the correlation to be better understood, more specific analysis is needed, such as identification of the bacterial genera to the species level. Moreover, bacterial communities often develop metal-resistant genes (MRGs) [83] in response to metal pollution. Hence, for future studies, identification of specific metal resistance genes, such as merA for Hg resistance [84]; cop A, cop B, pco A, pco C, and pco D for Cu resistance; and pbr T for Pb resistance [85] could be investigated to establish the functional capabilities of metal-resistant bacterial communities.

## 5. Conclusions

The objective of the study is to determine if bacterial bioindicators can serve as a tool for ecological risk assessment in a wetland ecosystem. To achieve this objective, microcosm systems with wetland sediments from the Pike River watershed were created to allow manipulations of pollutant types (e.g., nutrient and metals) and loading rates. These manipulations are not possible in a natural ecosystem and can help to determine specific patterns of bioindicator response to pollutant exposure at a regular interval. Metals (Pb, Cu) and nutrients ($NO_3^-$, $PO_4^{3-}$) were added to the microcosms at 7 day intervals. Bacterial DNA was extracted from the microcosm sediments, and taxonomical profiles of bacterial communities were identified up to the genera level. After analysis of the results, the following conclusions can be made concerning the research questions of this study:

The sediment bacterial indicators (Shannon and Simpson diversity indices of genera) were highly correlated with the pollutants, particularly the metals (Pb and Cu) added to the microcosms. This answers the first research question. We observed that the Cu added to the microcosm water negatively affected the bacterial diversity. The added Pb accumulated from the water column to the sediment of the microcosms, which increased the overall diversity of the bacterial community in relation to Pb. Hence, we observed some highly Pb-resistant genera in the microcosm sediments in the next part of the analysis where a specific assemblage of bioindicator bacterial genera in relation to metals such as Pb and Cu were identified. The genera *Ralstonia*, *Flavobacterium*, *Bacillus*, *Pseudomonas*, *Clostrodium*, *Aeromonas*, and *Arthrobacter* were identified as Pb-resistant, remediating, precipitating, and biomethylating by other studies in the literature and were also observed in the bacterial community identified within this study. The hierarchical cluster analysis in each metal microcosm showed the abundance level of each genus identified relative to their response to metals. In terms of abundance, genera such as *Fusibacter*, *Chlorobaculum*, *Prosthecobacter*, *Nocardioides*, *Aeromonas*, and *Arthrobacter* were identified as highly tolerant to the stress of Pb and Cu by being present throughout the experimental time in both low and high treatments of the microcosms relative to the controls. Genera such as *Ralstonia*, *Pseudomonas*, *Flavobacterium*, *Clostridium*, *Bacillus*, *Pedosphaera*, *Bdellovibrio*, *Holophaga*, and *Geobacter* were identified as tolerant (high or less). *Chlorobium* was identified as intolerant as this genus disappeared in the treatment microcosms relative to the controls as the experiment progressed.

**Author Contributions:** T.J.E. assisted in the study conceptualization, mentorship, design, data analysis, lab space, and funding of the project. C.F.W. helped in conceptualization, mentorship of the project, and manuscript edits. The microbial lab work was done in C.F.W.'s laboratory. S.A.M. assisted with sample collection, map creation, and manuscript edits. S.G.R. performed the conceptualization and sample collection. S.G.R. also planned and conducted all the lab work, data collection, data analysis, manuscript draft preparation, writing, reviewing, and manuscript edits. All authors have read and agreed to the published version of the manuscript.

**Funding:** The research was funded by The Village of Mount Pleasant (Racine, WI, USA) and the Wm. Collin Kohler's Foundation Sustainable Peacebuilding Fund at University of Wisconsin-Milwaukee.

**Institutional Review Board Statement:** Not Applicable.

**Informed Consent Statement:** Not Applicable.

**Data Availability Statement:** Land use data of the wetland sites is referred from https://www.sewrpc.org/SEWRPC.htm (accessed on 14 March 2015). The software that was used to analyze the bacterial data can be found from https://mothur.org/ (accessed on 14 March 2015).

**Acknowledgments:** We thank The Village of Mount Pleasant (Racine, WI, USA) and Wm. Collin Kohler's foundation for their support in this study. We are also greatly thankful to Erica Young, John Berges, and Neil O'Reilly (Department of Biological Sciences, University of Wisconsin-Milwaukee) for their constant support and feedback. A special thanks to Erica Young and John Berges for sharing laboratory space and equipment for the study.

**Conflicts of Interest:** The authors declare no conflict of interest.

## Appendix A

**Table A1.** Effect test from ANOVA showing the significance of experimental duration and treatment level (high and low) on metal (Pb and Cu) added in mg/L in the water and metal detected in ppm in the sediments of metal microcosm and nutrients ($NO_3^-$ and $PO_4^{3-}$) added in mg/L in the water of nutrient microcosm. * in the *p*-value column indicates a strong probability of significance.

| | Metal Microcosm—Metals Added in Water | | | | | |
| --- | --- | --- | --- | --- | --- | --- |
| | Pb (mg/L) | | | Cu (mg/L) | | |
| **Effect Source** | **SS** | **F Ratio** | *p*-**Value** | **SS** | **F Ratio** | *p*-**Value** |
| Experimental duration | 0.000001 | 24.763 | <0.0001 * | 0.0003 | 1.1749 | 0.319 |
| Treatment level | 0.000067 | 0.5031 | 0.61 | 0.0005 | 0.6749 | 0.515 |
| | Nutrient Microcosm—Nutrients Added in Water | | | | | |
| | $NO_3^-$ (mg/L) | | | $PO_4^{3-}$ (mg/L) | | |
| **Effect Source** | **SS** | **F Ratio** | *p*-**Value** | **SS** | **F Ratio** | *p*-**Value** |
| Experimental duration | 9067 | 6.4182 | 0.0036 * | 26 | 11.9591 | <0.0001 * |
| Treatment level | 79 | 0.0561 | NS | 6 | 2.5600 | NS |
| | Metal Microcosm—Metal Detected in Sediments | | | | | |
| | Ag (ppm) | | | As (ppm) | | | Cd (ppm) | | |
| **Effect Source** | **SS** | **F Ratio** | *p*-**Value** | **SS** | **F Ratio** | *p*-**Value** | **SS** | **F Ratio** | *p*-**Value** |
| Experimental duration | 0.0012 | 0.3312 | NS | 0.0767 | 4.2484 | 0.0223 * | 0.0054 | 0.4785 | NS |
| Treatment level | 0.0017 | 0.4522 | NS | 0.0307 | 1.7017 | NS | 0.0194 | 1.7162 | NS |
| | Fe (ppm) | | | Hg (ppm) | | | Ni (ppm) | | |
| **Effect Source** | **SS** | **F Ratio** | *p*-**Value** | **SS** | **F Ratio** | *p*-**Value** | **SS** | **F Ratio** | *p*-**Value** |
| Experimental duration | 0.0002 | 0.0096 | NS | 0.0540 | 2.7600 | NS | 0.000131 | 0.252300 | NS |
| Treatment level | 0.0001 | 0.0032 | NS | 0.0025 | 0.1261 | NS | 0.000078 | 0.149100 | NS |
| | Pb (ppm) | | | Rb (ppm) | | | | | |
| **Effect Source** | **SS** | **F Ratio** | *p*-**Value** | **SS** | **F Ratio** | *p*-**Value** | **SS** | **F Ratio** | *p*-**Value** |
| Experimental duration | 0.000011 | 1.9716 | NS | 0.0057 | 0.2966 | NS | 0.0473 | 0.3754 | NS |
| Treatment level | 0.000011 | 2.1328 | NS | 0.0083 | 0.4319 | NS | 0.0008 | 0.0067 | NS |

**Table A2.** (**A**) Number of sequences and the percent of total (more than 1% of total shown) for some detected phylum in nutrient and metal microcosms.

| (A) Phyla Detected | Number of Sequences for Each Phylum | Percent of Total |
| --- | --- | --- |
| Proteobacteria | 51,598 | 29.45% |
| Bacteroidetes | 23,252 | 13.27% |
| Chloroflexi | 16,236 | 9.27% |
| Planctomycetes | 10,013 | 5.71% |
| OD1 | 9477 | 5.41% |
| Firmicutes | 7513 | 4.29% |
| Acidobacteria | 7323 | 4.18% |
| Verrucomicrobia | 6934 | 3.96% |
| Actinobacteria | 5417 | 3.09% |
| Chlorobi | 3561 | 2.03% |
| Spirochaetes | 3428 | 1.96% |
| Armatimonadetes | 2702 | 1.54% |
| Elusimicrobia | 2148 | 1.23% |
| Cyanobacteria | 2099 | 1.20% |
| GN02 | 2090 | 1.19% |

**Table A3.** Effect test from ANOVA showing the significance of experimental duration and treatment level (high and low) on the Shannon and Simpson diversity indices of genera in nutrient and metal microcosm experiments. * in the *p*-value column indicates a strong probability of significance.

| | Shannon Diversity Index of Genera | | | Simpson Diversity Index of Genera | | |
|---|---|---|---|---|---|---|
| **Effect Source** | **SS** | **F Ratio** | **p-Value** | **SS** | **F Ratio** | **p-Value** |
| Experimental duration | 0.2765 | 3.3789 | NS | 1688.4 | 5.3548 | 0.0314 * |
| Treatment level | 0.0553 | 0.3376 | NS | 410.21 | 0.6505 | NS |
| | **Shannon Diversity Index of Genera** | | | **Simpson Diversity Index of Genera** | | |
| **Effect Source** | **SS** | **F Ratio** | **p-Value** | **SS** | **F Ratio** | **p-Value** |
| Experimental duration | 0.0804 | 14.0738 | 0.0013 * | 512 | 4.731 | 0.0418 * |
| Treatment level | 0.0143 | 1.2515 | NS | 93 | 0.4282 | NS |

**Table A4.** Multiple regression model with estimate (slope), combined $R^2$, and *p*-value (significance) of relationship between Shannon and Simpson diversity indices of genera and the metals detected (in ppm) in the sediments of metal microcosm.

| | Genera Shannon Diversity Index | | | Genera Simpson Diversity Index | | |
|---|---|---|---|---|---|---|
| **Parameter** | **Estimate** | **p-Value** | **$R^2$** | **Estimate** | **p-Value** | **$R^2$** |
| Ag(ppm) | 0 | NS | | 0 | NS | |
| As(ppm) | 0 | NS | | 0 | NS | |
| Cd(ppm) | 0 | NS | | 0 | NS | |
| Fe(ppm) | 0 | NS | | 0 | NS | |
| Hg(ppm) | 0 | NS | 0.00 | 0 | NS | 0.27 |
| Ni(ppm) | 0 | NS | | 0 | NS | |
| Pb(ppm) | 0 | NS | | −5606 | 0.0616 | |
| Rb(ppm) | 0 | NS | | 0 | NS | |
| Zn(ppm) | 0 | NS | | 57.85 | 0.0118 | |

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
