# Peer review of "Responses of Bacterial Taxonomical Diversity Indicators to Pollutant Loadings in Experimental Wetland Microcosms"

_water, doi:10.3390/w14020251_

Round 1

Reviewer 1 Report

Ms. Ref. No.: water-1526607
Title: Responses of bacterial taxonomical diversity indicators to pollutant loadings in experimental wetland microcosms
Journal: Water (ISSN 2073-4441)
Roy et al.

Review:

This study said it explores the pattern of bacterial responses to metal and nutrient pollution loading and seeks to evaluate whether bacterial indicators can be effective as a biomonitoring risk assessment tool for wetland ecosystems. It was conducted in laboratory microcosms by inoculating sediments collected from wetlands in an urbanizing watershed of Pike River, WI close to Lake Michigan. Bacterial community analysis was analysed with 16S bacterial rRNA sequencing.

The study found correlations between diversity indices of genera with metal and nutrient pollution as well as identified specific genera (including Fusibacter, Aeromonas, Arthrobacter, Bacillus, Bdellovibrio, and Chlorobium etc.). as predictive bioindicators for ecological risk assessment for metal pollution.

Concern: 

Introduction: Not enough, need to explain the question, justify hypothesis, present background studies and literature review, and write objectives.

Material and method: Please why you select these material collection sites. Was there any problem on the site?

Result: Please rewrite the result. It is unclear. You have not presented your figure and tables properly. Presenting figure and tables are not enough, you need to create a story of it.

Figures: Figures quality is very poor. Please improve the quality of the figures.

Discussion: What is your finding? compare your finding with earlier related studies, and explain why you get this result? why not the next result.

Conclusion: Need to write again. Language structures are not scientific.

Appendix: I think scientific papers use Supplemental materials, not an appendix.

Overall: The paper is written very poorly. Very unclear message. No-story. Need a major revision

Suggestion: Take one published paper. Follow a similar way how it has written.  Show your manuscript to your senior authors before submission. Please request them for reading the manuscript more carefully. Otherwise, it is just a time waste of reviewer volunteers.

ISSUE 1: Line 29: Can you avoid two such as in a single sentence?

ISSUE 2: Line 29-31: More relevant citation- https://doi.org/10.1186/s40793-021-00379-w

ISSUE 3: Line 32: Multiple pathways such as ……

ISSUE 4: Line 34-36: what are these indicators..

Issue 5: Line 169- why appendix table- is it Supplemental table? Please change everywhere appendix- Supplemental

Issue 6: Line 171- p < 0.0001 ?

Issue 7: Figure 7 quality is so poor. Make the Y-axis title more clear.

Issue 8: Figure 8A- what is the x-axis- Please make it clear.

Issue: 9: Line- 436-437- are you connecting medians with lines or mean, in boxplot?  Better to connect the median.

ISSUE 10: Line 441-453: Are these mistakenly or purposefully here?

Author Response

Cover letter

Thank you for reviewing our paper. Please find below the answers to your questions. These edits have been incorporated into the paper as well. Please note that we encountered a small problem with the “Track changes” and the “line number” working together. We have fixed the issue. But if the problem persists, please refer to the page numbers included below.

Introduction: Not enough, need to explain the question, justify hypothesis, present background studies and literature review, and write objectives.

All these changes have been made between lines 31-63, pages 1-2, or in the Introduction section

Material and method: Please why you select these material collection sites. Was there any problem on the site?  

The reason the sites were chosen is now included. See lines 70-75, page 2-3

Result: Please rewrite the result. It is unclear. You have not presented your figure and tables properly. Presenting figure and tables are not enough, you need to create a story of it.

A major part of the result is now re-written. The story has been written now as bacterial indicators being the central character and the steps have been specified to reach the conclusion.

Figures: Figures quality is very poor. Please improve the quality of the figures.

All the figures and their axes have been redone

Discussion: What is your finding? compare your finding with earlier related studies, and explain why you get this result? why not the next result.

A Major part of the discussion is now re-written. More literature supporting the results and mechanisms of actions are elaborated. See in lines: 345-360, 371-376, 379-396, 428-437 or in pages 14,15.

Conclusion: Need to write again. Language structures are not scientific.

The conclusion is now rewritten, with more illustration and specific findings. See page 16

Appendix: I think scientific papers use Supplemental materials, not an appendix.

Appendices replaced with Supplementary tables

ISSUE 1: Line 29: Can you avoid two such as in a single sentence?

Sentence broken

ISSUE 2: Line 29-31: More relevant citation- https://doi.org/10.1186/s40793-021-00379-w

Relevant citation added.

ISSUE 3: Line 32: Multiple pathways such as ……

Pathway specified (line 39-40) or page 1

ISSUE 4: Line 34-36: what are these indicators..

indicator specified (line 35)or page 1

Issue 5: Line 169- why appendix table- is it Supplemental table? Please change everywhere appendix- Supplemental : 

All changed to supplemental

Issue 6: Line 171- p < 0.0001 ?

I am a little confused by this question. P indicates P-value. The change applied in line 190 or page 6

Issue 7: Figure 7 quality is so poor. Make the Y-axis title more clear.

The figure number has changed to 8. The Y-axis has been specified.

Issue 8: Figure 8A- what is the x-axis- Please make it clear.

The figure number has changed to 9A. The x-axis has been specified.

Issue: 9: Line- 436-437- are you connecting medians with lines or mean, in boxplot?  Better to connect the median.

This figure is no longer included

ISSUE 10: Line 441-453: Are these mistakenly or purposefully here?

The templates are removed

Reviewer 2 Report

See attached file

Author Response

Thank you for reviewing our paper. Please find below the answers to your questions. These edits have been incorporated into the paper as well. Please note that we encountered a small problem with the “Track changes” and the “line number” working together. We have fixed the issue. But if the problem persists, please refer to the page numbers included below.

-If available, please include/discuss physico-chemical characteristics: main contaminants/discharge volume etc, from the waste water discharges

More site characteristics are included in Table 1. See page 3. Also, why these wetlands were chosen are discussed in lines 70-75(page 2). That explains the discharges in the wetland sites.

Now as this is a microcosm study, the contaminants, volume of the microcosm, etc. are already specified in Figure 4, Table 2 (page 5). A picture of the microcosm built is now also specified in Figure 3 (page 4)

-The results and discussion chapter requires re-writing and supplementing with specific own or literature results. Please indicate references for aquatic systems contamination examples throughout the world. I suggest to read:

Pradit, S., Wattayakorn, G., Angsupanich, S., Baeyens, W., Leermakers, M. (2010). Distribution of Trace Elements in Sediments and Biota of Songkhla Lake, Southern Thailand. Water, Air, & Soil Pollution, 206:155–174 DOI 10.1007/s11270-009-0093-x.

Prokisch, J., Szeles, E., Kovacs, B., Gyori, Z., Nemeth, T., West, L. (2009). Sampling Strategies for Testing and Evaluation of Soil Contamination in Riparian Systems at the Tisza River Basin, Hungary. Communications in Soil Science and Plant Analysis, 40:391–406.

Jiménez-Ballesta R. García-Navarro F.J, Bravo S., Amoros JA, Pérez de los Reyes C. and Mejias M. (2017). Environmental assessment of potential toxic elements contents in the inundated floodplain área of Tablas de Daimiel wetland (Spain).  Environ Geochem and Health 39: 1159-1177. DOI:10.1007/s10653-016-9884-3.

A major part of the discussion is now re-written. With more literature supporting the results and mechanisms of actions are elaborated. These references are now included. See in line 345-360, 371-376, 379-396, 428-437 or in pages 14,15.

-I suggest to make a connection of the analysis and the causes of degradation.

These mechanisms of actions are elaborated in the lines 345-360, 371-376, 379-396, 428-437, supported with literature or in pages 14,15.

-Please, I suggest to to rewrite the conclusions.

The conclusion is now rewritten, with more illustration and specific findings. See page 16

-In the Referenece section please remove the reference models 

templates removed

Reviewer 3 Report

  1. Add some results (in percentage) in the abstract.
  2. Language should be also be corrected.

The introduction is not having latest references of 2018, 2019 and 2020.

Results are well written. However, every parameter should be written as increase or decrease in percentage and the results should be comprehensive (in detailed). Here the readers will get nothing. So revise the result section.

Discussion part is very week. The mechanism of action in the discussion part is not well documented. I suggest the authors should add mechanism of action, why and how a parameter is decreased or increased, what is the real mechanism?. It is important. Also authors should support their results by already published recent articles.

Author Response

Thank you for reviewing our paper. Please find below the answers to your questions. These edits have been incorporated into the paper as well. Please note that we encountered a small problem with the “Track changes” and the “line number” working together. We have fixed the issue. But if the problem persists, please refer to the page numbers included below.

  1. Add some results (in percentage) in the abstract.

percentages added. See in lines 190-210 (page 6).

  1. Language should be also be corrected.

Language and grammatical changes have been applied throughout the paper

The introduction is not having latest references of 2018, 2019 and 2020.

Latest references included in the introduction. See reference number: 2,9,10,11,16,17,27,28

Results are well written. However, every parameter should be written as increase or decrease in percentage and the results should be comprehensive (in detailed). Here the readers will get nothing. So revise the result section.

The results are in detail now. Percentages for increase/decrease added. See in lines 190-210 (page 6).

Discussion part is very week. The mechanism of action in the discussion part is not well documented. I suggest the authors should add mechanism of action, why and how a parameter is decreased or increased, what is the real mechanism?. It is important. Also authors should support their results by already published recent articles.

A major part of the discussion is now re-written. More literature supporting the results and mechanisms of actions are elaborated. These references are now included. See in line 345-360, 371-376, 379-396, 428-437 or in pages 14,15.

Reviewer 4 Report

This manuscript is well written and organized to explore the response of bacterial taxonomical diversity indicators to metal and nutrient pollutants loading in the wetland microcosms. Some minor comments are given as below.

  1. Line 53, “from [9][10]” would be “from [9,10]”.
  2. If possible the picture for microcosms can be supplied with figure.
  3. Line 90, “dissolved in RO water” would be “dissolved in reverse osmosis (RO) water.” And Line 92, “dissolved in reverse osmosis (RO) water” can be “dissolved in RO water”
  4. Figure 7 is not clear. It should be revised with clear one.
  5. Line 240, “Figure 8-9” can be revised as “Figures 8 and 9”.
  6. Figure 8A. Wetland site 1 m microcosm is also not clear. It can be replaced with clear one.
  7. In the Conclusions, the section is too short to be extended.
  8. In the “References”, No. 1 to No. 8 should be deleted, because they are template.

Author Response

Thank you for reviewing our paper. Please find below the answers to your questions. These edits have been incorporated into the paper as well. Please note that we encountered a small problem with the “Track changes” and the “line number” working together. We have fixed the issue. But if the problem persists, please refer to the page numbers included below.

  1. Line 53, “from [9][10]” would be “from [9,10]”.

Change applied. See line 67 in page 2. (Ref number changed)

2. If possible the picture for microcosms can be supplied with figure.

See Figure 3

3. Line 90, “dissolved in RO water” would be “dissolved in reverse osmosis (RO) water.”

And Line 92, “dissolved in reverse osmosis (RO) water” can be “dissolved in RO water”

Change applied in line 81(page 3)

4. Figure 7 is not clear. It should be revised with clear one. Figure replaced

5. Line 240, “Figure 8-9” can be revised as “Figures 8 and 9”.

Change applied. See line 285 page 10.

6. Figure 8A. Wetland site 1 m microcosm is also not clear. It can be replaced with clear one. Figure replaced

  1. In the Conclusions, the section is too short to be extended.

The conclusion is now rewritten, with more illustration and specific findings. See page 16

8. In the “References”, No. 1 to No. 8 should be deleted, because they are template.

templates removed

Round 2

Reviewer 1 Report

The authors had improved the quality of the manuscript. 

Reviewer 2 Report

The authors have followed most of the recommendations given. Therefore, as a consequence of the changes and corrections incorporated in this new version, the scientific quality of the manuscript has considerably improved. Then, it can be accepted.

Reviewer 4 Report

The authors have well responded my previous comments. I recommend that the revised manuscript can be accepted for publication in “Water” Journal.